# Metabolomic analysis of the impact of MtrA on carbon metabolism in *Streptomyces coelicolor*

Yanping Zhu,[1,2] Hanlei Zhang,[1] Xiuhua Pang[1]

**ABSTRACT** The response regulator MtrA of *Streptomyces* regulates secondary metabolism as well as primary metabolism, including nitrogen metabolism and phosphate metabolism; however, it is not known whether MtrA is involved in the control of central carbon metabolism in *Streptomyces*. In this study, we revealed that the growth medium of the MtrA mutant strain (Δ*mtrA*) is acidic under multiple growth conditions and that this acidification is dependent on the type of medium used. We performed targeted metabolomic analysis to determine the types and levels of organic acids produced by the wild-type strain *Streptomyces coelicolor* M145 and Δ*mtrA*, and the results revealed that production of multiple organic acids associated with the tricarboxylic acid cycle (TCA) and glycolysis pathway was changed significantly in Δ*mtrA*, compared with M145, indicating a broad impact of MtrA on carbon metabolism and suggesting the basis for the acidification of the growth media by Δ*mtrA*. Multiple potential MtrA sites were predicted in the sequences upstream of genes involved in the TCA cycle, including genes encoding citrate synthases, and we showed that MtrA bound these potential sites, suggesting that MtrA targets these carbon metabolism genes. Our transcriptional analysis showed that carbon metabolism genes with MtrA sites are differentially expressed in Δ*mtrA*, indicating regulation of these genes by MtrA. Overall, our study indicates that the response regulator MtrA has a broad impact on central carbon metabolism, adding new insight into our understanding of the regulation of carbon metabolism in *Streptomyces*.

**IMPORTANCE** Central carbon metabolism is a key primary metabolic process, and its tight regulation is crucial for maintaining normal physiology in microbes. However, carbon metabolism is the least understood metabolic process in primary metabolism in *Streptomyces*. In this study, we demonstrated a broad impact of the response regulator MtrA on the production of metabolites associated with the tricarboxylic acid cycle and glycolysis pathway, thereby leading to the accumulation of organic acids and decreases in the pH values of the growth medium with an MtrA mutant strain. We further revealed MtrA sites upstream of genes involved in carbon metabolism and determined that MtrA bound to these sites, revealing MtrA as a regulator for carbon metabolism in *Streptomyces*. Our study enhances the understanding of the role of MtrA and helps to elucidate the regulatory mechanisms of a major metabolic process in *Streptomyces*.

**KEYWORDS** *Streptomyces*, carbon metabolism, organic acid, MtrA

*S*treptomyces species are soil-dwelling, Gram-positive, filamentous bacteria. Two outstanding features enable these microbes to stand out from other microbes. *Streptomyces* species have a relatively complex lifecycle involving spore formation, and they can differentiate into several cell types, including vegetative hyphae, aerial hyphae, and spores, depending on the developmental stage (1–3). *Streptomyces* species also have great potential for producing metabolites for clinical applications, including antibiotics,

**Peer Reviewer** Guoqing Niu, Southwest University, Chongqing, China

Address correspondence to Xiuhua Pang, pangxiuhua@sdu.edu.cn.

The authors declare no conflict of interest.

and are generally recognized as a vast reservoir for discovering new compounds of clinical importance; *Streptomyces* produce about two-thirds of known antibiotics (2), which are generally metabolites produced at the late stages of cell growth and are the so-called secondary metabolites.

In *Streptomyces*, cellular development and secondary metabolism are controlled by many regulatory factors, directly or indirectly, with some factors impacting both development and secondary metabolism (1, 4). One of the known regulators with such a role is MtrA, which is a conserved response regulator of a two-component signal transduction system (TCS) MtrAB in actinobacteria (5). MtrA of *Streptomyces coelicolor* has high amino acid sequence similarity to MtrA (MtrA$_{MTB}$) of *Mycobacterium tuberculosis*, which is an essential gene in that species (6, 7), and MtrA (MtrA$_{CGL}$) of *Corynebacterium glutamicum*, which is required for the regulation of cell wall metabolism and osmoprotection (8). In *Streptomyces*, it has been shown that MtrA functions as a regulator in antibiotic production, repressing production of actinorhodin and undecylprodigiosin but activating production of the calcium-dependent antibiotic and yellow-pigmented type-I polyketide (9–11); MtrA also functions as a developmental regulator, controlling development-associated genes, including developmental regulatory genes, such as *bld* and *whi*, and genes involved in the formation of the hydrophobic sheath, such as *chp* and *rdl* (10, 12). In addition to the control of secondary metabolism and development, MtrA controls primary metabolism, including phosphate metabolism, in which MtrA targets the main phosphate metabolism gene *phoA* and the principal regulator of phosphate metabolism, *phoP* (13); and nitrogen metabolism (14). Interestingly, MtrA recognizes the DNA binding sequence of GlnR, the primary regulator of nitrogen metabolism in *Streptomyces* (15, 16), and competes for binding to GlnR sites (14). However, in nitrogen metabolism, MtrA mainly functions as a repressor for nitrogen metabolism genes under nutrient-rich conditions, enabling the bacteria to avoid the unnecessary assimilation of inorganic nitrogen sources under these conditions, whereas GlnR activates nitrogen metabolism genes under nitrogen-limited (nutrient-poor) conditions (15, 16), and thus these two regulators coordinate the regulation of nitrogen metabolism under different growth conditions (17). The genome of *S. coelicolor* encodes more than 60 paired TCSs (18). Similar to MtrA, characterized response regulators of *Streptomyces*, such as PhoP (19), AfsQ1 (20), and DraR (21), have been demonstrated to have critical roles in modulating the physiology of the host strain (22).

Carbon metabolism is the most important primary metabolism in nearly all forms of life. Central carbon metabolic pathways, such as glycolysis, tricarboxylic acid (TCA) cycle and pentose phosphate pathway, and gluconeogenesis, are present in *Streptomyces* (18). However, multiple gene copies encoding similar or homologous gene products associated with these pathways are a common feature in *Streptomyces*; for example, *S. coelicolor* has four copies of genes encoding citrate synthases, three copies of genes encoding 6-phosphofructokinases, and three copies of genes encoding 6-phosphogluconate dehydrogenases (18), and usually, these multiple copies of central carbon metabolic genes are scattered in the *Streptomyces* genome, enabling coordinated regulation of these genes a difficult task. Additionally, compared with nitrogen and phosphate metabolism, control of carbon metabolism has been relatively less investigated. However, DasR has been characterized as a regulator for carbon metabolism in *Streptomyces* (23), responding to glucosamine-6-phosphate and controlling a regulon that includes an N-acetylglucosamine (monomer of chitin) transporter and genes involved in the metabolism of N-acetylglucosamine (23). In this study, we discovered that the regulatory repertoire of MtrA includes carbon metabolism, providing new insights into the role of this major regulator in the regulation of carbon metabolism in *Streptomyces*.

## RESULTS AND DISCUSSION

### Acidification of the growth medium by the *S. coelicolor* mutant Δ*mtrA*

We previously found that the *S. coelicolor mtrA* deletion mutant strain (Δ*mtrA*) grows poorly on solid media such as YBP (11, 12). To investigate the potential factors that might impact the growth of this mutant, the pH value of the growth medium for Δ*mtrA* and the control strain M145 was measured on YBP (Fig. 1; Fig. S1). Although little difference was observed between the first two time points of 24 and 36 h for the pH value (close to 7.0) of the M145 growth medium, the pH value increased steadily after 36 h for this strain, reaching about pH 8.6 at 96 h (Fig. 1). However, the pH value of the Δ*mtrA* growth medium decreased abruptly, reaching pH 5.0, about two full pH points lower than for M145 at the same time, and then the pH value of the Δ*mtrA* growth medium increased steadily from 36 h, reaching a peak value at 84 h before dropping slightly (Fig. 1). The growth medium of the *mtrA*-complemented strain C-Δ*mtrA* displayed a temporal pH pattern nearly identical to that of M145 (Fig. 1). Overall, the pH value for Δ*mtrA* was at least one full pH point lower than for M145 at each time point, indicating acidification of the YBP growth medium by Δ*mtrA*. Acidification of the growth medium by Δ*mtrA* was also revealed on the complex MS medium (Fig. S2), which has a very different composition from YBP (24), supporting growth medium acidification by Δ*mtrA* under different growth conditions.

YBP is a complex medium mainly composed of yeast extract, beef extract, peptone, and glucose, in addition to other defined components (25). To investigate whether Δ*mtrA* could still acidify a less complex growth medium, M145 and Δ*mtrA* were grown on solid R2YE (Fig. S3), which contains yeast extract in addition to other defined components (24). Very little shift in pH value was observed with M145 grown on R2YE from the first (24 h) to the last (144 h) time point (Fig. 1). In contrast, the pH value of the Δ*mtrA* growth medium decreased steadily from 24 to 48 h, reaching about pH 6.0, and then the pH value remained relatively steady with only a slight additional decrease through

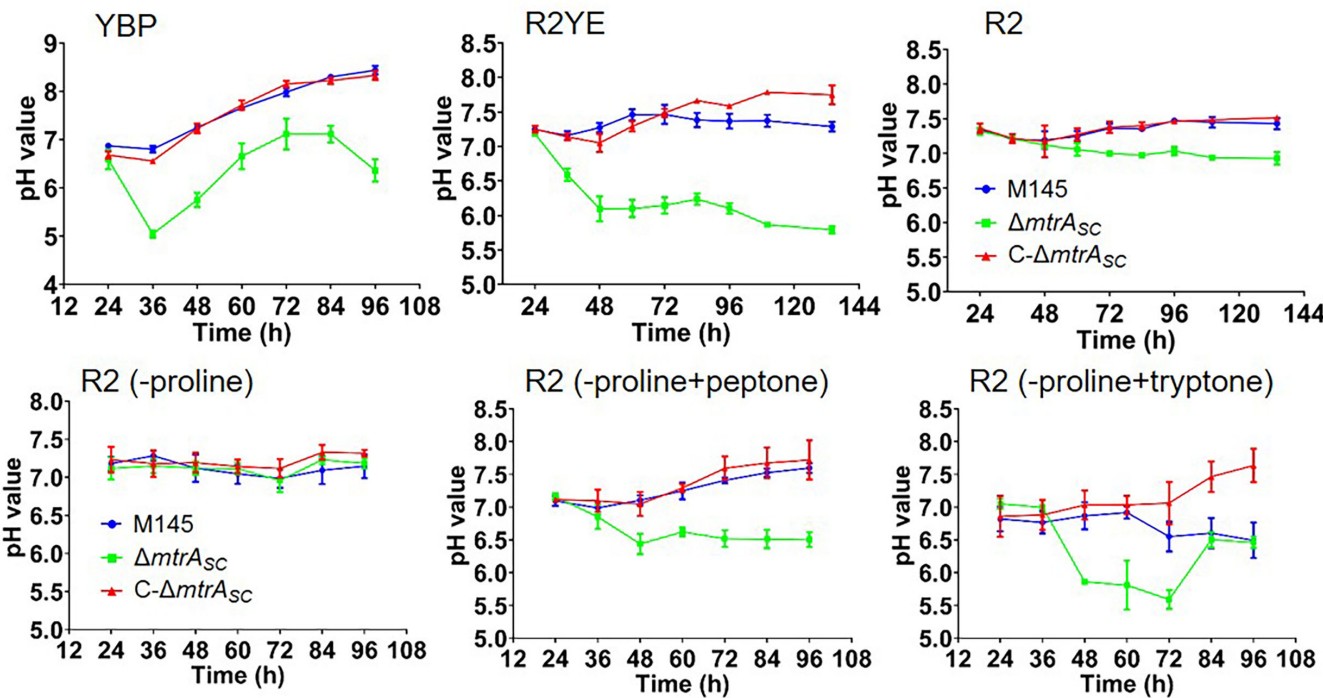

**FIG 1** Temporal pH values of the growth medium for the *mtrA* mutant strain of *S. coelicolor* under different growth conditions. The pH values of the growth medium were measured for the wild-type strain *S. coelicolor* M145, the *mtrA* deletion mutant Δ*mtrA*, and the *mtrA*-complemented strain C-Δ*mtrA* grown on YBP, R2YE, R2, and modified R2 without (−) proline, and with (+) added peptone or tryptone as indicated.

the last time points. The complemented strain C-ΔmtrA displayed a temporal pH pattern similar to that of M145 (Fig. 1). Collectively, the data showed a difference of at least one full pH point between M145 and ΔmtrA at 48 h and time points afterward, indicating acidification of the growth medium by ΔmtrA when using R2YE.

Yeast extract is a mixture of proteins and carbohydrates that can potentially lead to a decrease in pH, and therefore, to investigate the contribution of yeast extract on the acidification of R2YE by ΔmtrA, we tested the pH value of ΔmtrA grown on R2 (Fig. S4), which has the same components as R2YE, except without the addition of yeast extract (24). The temporal pH curves were very flat for both M145 and ΔmtrA grown on R2, and the difference was reduced to less than 0.5 in pH value (Fig. 1), suggesting that yeast extract is a nutrient factor contributing to the acidification by ΔmtrA. We also evaluated the effects of proline, which is a component of R2 and a glucogenic amino acid whose metabolism can result in the formation of α-ketoglutarate, an intermediate of the tricarboxylic acid cycle, potentially leading to acidification of the growth medium. Elimination of proline in R2 completely abrogated the difference in pH value between M145 and ΔmtrA, whereas replacement of proline by a nitrogen source in a more complex form, such as peptone (Fig. S5) or tryptone, enlarged the difference in pH value of the growth medium between M145 and ΔmtrA (Fig. 1). It appears that the more complex the nitrogen source, the lower the resulting pH. It is possible that more complex nitrogen sources, such as yeast extract, tryptone, and peptone, may result in the generation of more glucogenic amino acids whose metabolism would lead to the production of more organic acids, thereby resulting in a lower pH (Fig. 1). The ability of *Streptomyces* species to excrete organic acids has been reported (26, 27), consistent with the acidification of the medium.

In addition, we showed that the Tris(hydroxymethyl)methyl-2-aminoethanesulphonic acid (TES) component in R2YE helped to buffer the pH of the ΔmtrA medium; the pH value with ΔmtrA decreased to about pH 5.0 or lower on R2YE without the addition of TES (Fig. S6), further magnifying the difference in pH value between M145 and ΔmtrA. Collectively, our data indicate that the MtrA-dependent acidification of growth medium is nitrogen reagent-dependent, which is in agreement with the role of MtrA as a repressor for nitrogen metabolism genes on nutrient-rich conditions (14). In contrast, the *glnR* mutant (ΔglnR) of *S. coelicolor* did not acidify its growth medium when grown on nutrient-rich media such as YBP and R2YE (Fig. S7). GlnR is a nitrogen metabolism activator on nutrient-poor (nitrogen-limited) medium, whereas under nutrient-rich conditions, *glnR* is repressed by MtrA (14–16, 28); therefore, the absence of *glnR* was not expected to affect pH during growth on YBP and R2YE, consistent with our observations and further supporting differences in the regulatory roles of GlnR and MtrA.

In general, a neutral pH favors growth, and an acidic pH negatively affects the growth of microbes (29–31). Mutation of *citA*, encoding the major citrate synthase that catalyzes the synthesis of citrate by oxaloacetate and acyl-CoA, or *acoA*, encoding an aconitase that catalyzes the reversible isomerization of citrate to isocitrate, in *S. coelicolor* led to the accumulation of organic acids and a marked decrease in the pH value of the growth medium (26, 31), as did deletion of *mtrA*. Additionally, mutation of *cya*, which encodes an enzyme for the synthesis of cyclic adenosine monophosphate (AMP), led to the accumulation of organic acids by *S. coelicolor* under specific growth conditions (32). Mutant strains of these genes are also developmentally blocked, forming a bald phenotype (26, 31, 32), similar to that of ΔmtrA grown on YBP and R2YE (Fig. S1 and S3). However, the morphological defects of the *citA* and *cya* mutants could be rescued by the addition of pH buffer (26, 32), suggesting that an acidic pH was responsible for the poor growth and development of these mutant strains. In contrast, although the pH value of the growth medium of ΔmtrA was lower than that of the wild-type strain M145 under most conditions, and this low pH may potentially retard the growth of ΔmtrA, the low pH was not responsible for the bald phenotype of ΔmtrA, as ΔmtrA still displayed a bald phenotype on R2 (Fig. S4), where the growth medium of both M145 and ΔmtrA had a neutral pH (Fig. 1). Therefore, our results indicate that the bald phenotype of

Δ*mtrA* is not caused by an acidic pH value; however, the acidification of growth media by Δ*mtrA* is likely caused by the accumulation of organic acids, similar to the *citA* and *acoA* mutants. Notably, mutants of the developmental genes *bldA*, *bldB*, *bldC*, *bldD*, and *bldG* also irreversibly acidified their growth medium, and as with Δ*mtrA,* these bald mutants were not suppressed by neutralizing buffer (32); these findings indicate a similar phenotype (i.e., a pH-unrelated bald phenotype) among mutants of genes associated with development.

## Acidification of the growth medium by Δ*mtrA* of *S. lividans* and *S. venezuelae*

To investigate if MtrA orthologues have a similar effect on media acidification, we tested the pH using *Streptomyces lividans* Δ*mtrA* (Δ$mtrA_{SLI}$) grown on solid YBP and R2YE, which were markedly acidified by *S. coelicolor* Δ*mtrA* (Fig. 1). With wild-type *S. lividans* strain 1326, YBP maintained a relatively constant pH value (about pH 7.2) from 24 to 108 h, whereas the pH level with the mutant strain Δ$mtrA_{SLI}$ decreased by 36 h and reached the lowest value at 72 h (pH 5.5) (Fig. S8A); although the pH value increased slightly by 84 h and afterward with Δ$mtrA_{SLI}$, the difference in pH value was always more than 1.0 between the wild-type 1326 and Δ$mtrA_{SLI}$ from 36 h onward (Fig. S8A). The growth medium of the complemented strain C-Δ$mtrA_{SLI}$ demonstrated a temporal pH pattern similar to that observed with the wild-type strain (Fig. S8A). For R2YE medium, a lower pH value was consistently detected from 24 h onward for the mutant strain Δ$mtrA_{SLI}$, compared with the wild-type strain 1326, with the peak difference in pH value (about 2.2 pH points) measured at 72 h (Fig. S8A). With *Streptomyces venezuelae* Δ*mtrA*, the pH value of the YBP and R2YE growth media was also lower than with the control strain at most of the time points (Fig. S8B). Collectively, our data indicated that Δ*mtrA* mutants of *Streptomyces* species other than *S. coelicolor* can also acidify their growth medium, suggesting a conserved role of MtrA in the production of organic acids.

## Production of organic acids by Δ*mtrA* on solid YBP

Growth medium acidification was indicative of the accumulation of organic acids by the mutant strain Δ*mtrA*. To identify the organic acids that potentially led to the change of pH, we performed targeted metabolomic analysis to quantify the levels of metabolites related to carbon metabolism, including metabolites from the TCA cycle and glycolysis pathway, which mainly produce organic acids. We first compared the metabolomic data of M145 and Δ*mtrA* using samples grown on YBP for 36 h, as a large difference in pH value (about 2.0 pH points) was observed between M145 and Δ*mtrA* at this time (Fig. 1).

Significantly higher levels of alpha-ketoglutarate, succinate, fumarate, and malic acid were produced in Δ*mtrA* than in M145 (Fig. 2A); although the levels of citrate, cis-aconitate, and isocitrate appeared higher in Δ*mtrA* than in M145, the differences were not statistically significant (Fig. S9). However, the levels of metabolites from the glycolysis pathway, including glucose 6-phosphate, fructose 6-phosphate, fructose 1,6-bisphosphate, and 3-phosphoglycerate, were consistently significantly higher in Δ*mtrA* than in M145 (Fig. 2B). These data indicated a broad impact of MtrA on metabolites of carbon metabolism, with a high level of accumulation of multiple organic acids in Δ*mtrA*, which likely caused the acidification of the growth medium by Δ*mtrA*.

## Production of organic acids by Δ*mtrA* on solid R2YE

To further track the dynamics of metabolites associated with the TCA cycle and glycolysis pathway, we then performed metabolomic analysis using samples of M145 and Δ*mtrA* grown on R2YE for 36 h, when the pH value of the Δ*mtrA* medium started to decrease (Fig. 1), and at 48 h, when the pH began to temporarily stabilize. At 36 h, the levels of organic acids in the TCA cycle, including citrate, cis-aconitate, isocitrate, alpha-ketoglutarate, succinate, and fumarate, were significantly higher in M145 than in Δ*mtrA* (Fig. 2 and 3); the only exception was oxaloacetate, whose concentration was the highest among all metabolites detected at 36 h in Δ*mtrA* and whose level was significantly higher in Δ*mtrA*

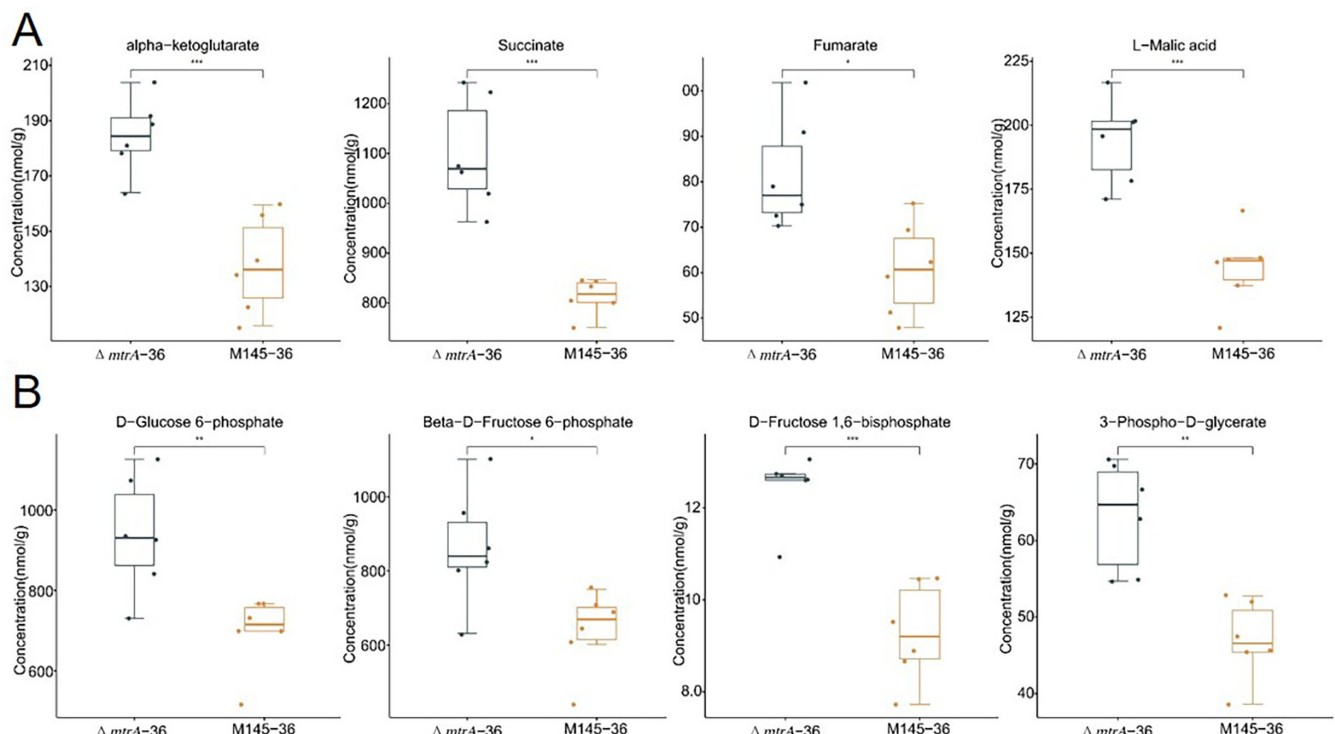

**FIG 2** Quantification of carbon metabolism metabolite production in *S. coelicolor* on YBP. Quantification of metabolites from the (A) TCA cycle and (B) glycolysis pathway. *S. coelicolor* wild-type strain M145 and Δ*mtrA* were grown at 30°C on solid YBP medium for 36 h; the boxplot data were obtained using six different sample preparations. Student's *t*-test was used for comparison; *$P < 0.05$, **$P < 0.01$, ***$P < 0.005$.

than in M145 (Fig. 4A). Although the level of malic acid appeared higher in Δ*mtrA* than in M145, the difference was not significant (Fig. S10). For metabolites of the glycolysis pathway, the levels of glucose 6-phosphate, pyruvate, and its derivative, lactate, were significantly higher in M145 than in Δ*mtrA* at 36 h, whereas the levels of fructose 6-phosphate, fructose 1,6-bisphosphate, and 3-phosphoglycerate were significantly higher in Δ*mtrA* than in M145 (Fig. S11). Given the lower level of multiple organic acids in Δ*mtrA* than in M145 and the acidic pH already present with Δ*mtrA* at 36 h, we deduced that the accumulation of oxaloacetate contributed to the acidic pH at this time point (Fig. 1 and 4A).

Next, we compared the levels of carbon metabolites at 48 h of growth on R2YE. Overall, the trends in metabolite differences between M145 and Δ*mtrA* observed at 36 h were maintained at 48 h (Fig. 4B and 5; Fig. S12) except that no significant difference was detected for succinate at 48 h (Fig. S10). Additionally, no significant difference was detected for glucose 6-phosphate between M145 and Δ*mtrA*, and while the level of pyruvate was significantly lower in Δ*mtrA* than in M145, the level of lactate, the derivative of pyruvate, was significantly higher in Δ*mtrA* than in M145 (Fig. S12), potentially due to the greater expression of a lactate dehydrogenase that converts the pyruvate to lactate. Although the presence of a lactate dehydrogenase has not been confirmed in *S. coelicolor*, SCO3594 is annotated as a putative D-lactate dehydrogenase and may play a role here. Collectively, our data showed again that MtrA has a broad impact on the production of the metabolites from the TCA cycle and glycolysis pathway, reflecting a major role for MtrA in metabolic control.

## Metabolomic analysis reveals dynamics of carbon metabolites

We noticed that, while the concentration of oxaloacetate at 36 and 48 h in M145 was comparable (Fig. 4C), the concentration of oxaloacetate at 48 h was significantly higher than at 36 h in Δ*mtrA* (Fig. 4D), suggesting more accumulation of oxaloacetate in Δ*mtrA*

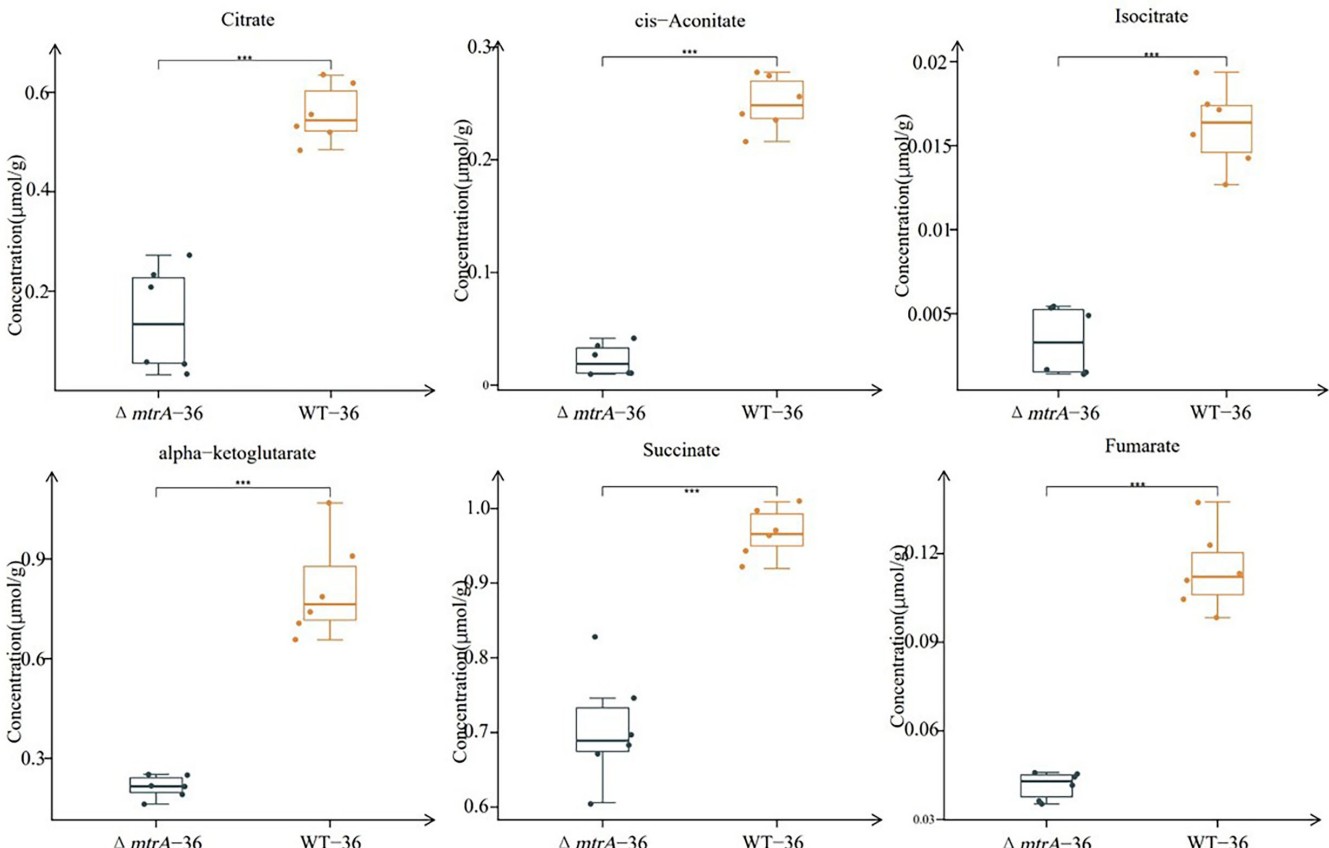

**FIG 3** Quantification of TCA cycle metabolite production on R2YE at 36 h. *S. coelicolor* wild-type strain M145 and Δ*mtrA* were grown at 30°C on solid R2YE medium for 36 h; the boxplot data were obtained using six different sample preparations. Student's *t*-test was used for comparison; *P < 0.05, **P < 0.01, ***P < 0.005.

and potentially explaining the lower acidic pH value at 48 h (Fig. 1). To have a better insight into the dynamics of key carbon metabolites, we compared the level of these metabolites at 36 and 48 h in M145 and in Δ*mtrA*. According to the observed patterns, these metabolites could be divided into three groups. Group 1 metabolites demonstrated a similar trend in M145 and Δ*mtrA*; this group includes isocitrate (Fig. S13), fumarate, and malic acid (Fig. S14 and S15). Therefore, fumarate and malic acid would not contribute to the decrease in pH value from 36 to 48 h in the mutant. Group 2 metabolites had similar concentrations at 36 and 48 h in Δ*mtrA*, showing no significant change in the

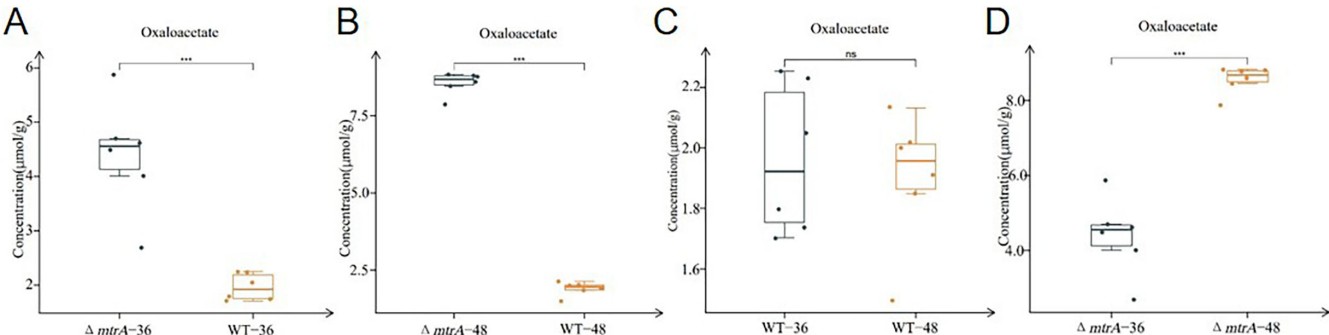

**FIG 4** Quantification of oxaloacetate production. Comparison of oxaloacetate production between (A, B) Δ*mtrA* and M145 at (A) 36 h and (B) 48 h; and (C, D) 36 h and 48 h in (C) M145 and (D) Δ*mtrA*. The *S. coelicolor* wild-type strain M145 and Δ*mtrA* were grown at 30°C on solid R2YE medium; the boxplot data were obtained using six different sample preparations. Student's *t*-test was used for comparison; *P < 0.05, **P < 0.01, ***P < 0.005; ns, not significant.

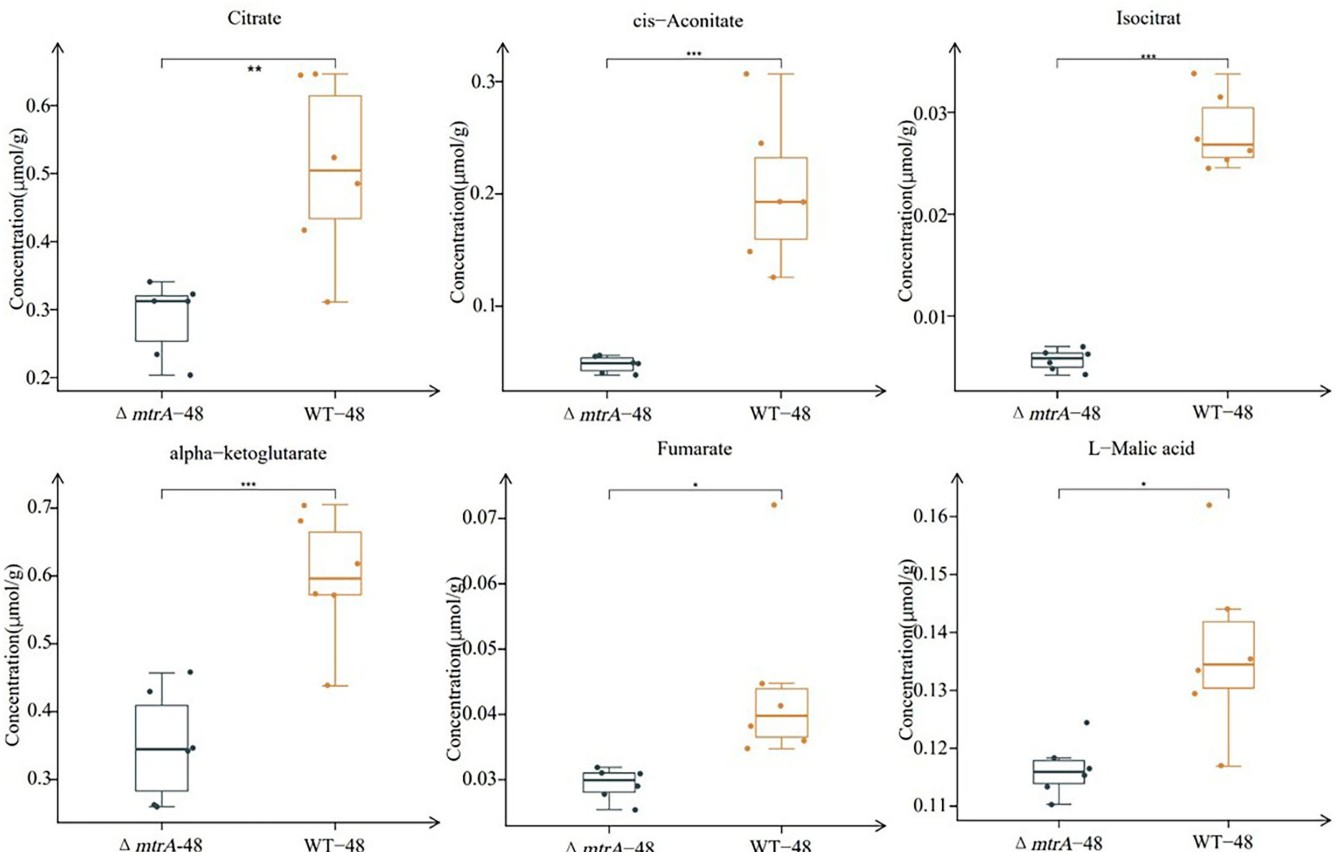

**FIG 5** Quantification of TCA cycle metabolite production on R2YE at 48 h. *S. coelicolor* wild-type strain M145 and Δ*mtrA* were grown at 30°C on solid R2YE medium for 48 h; the boxplot data were obtained using six different sample preparations. Student's *t*-test was used for comparison, *$P < 0.05$, **$P < 0.01$, ***$P < 0.005$.

mutant, regardless of their difference in M145. This group includes pyruvate and succinate (Fig. S16 and S17), these two organic acids also did not contribute to the decrease in pH value from 36 to 48 h with strain Δ*mtrA*. Group 3 metabolites had higher concentrations at 48 h than at 36 h in Δ*mtrA*; this group includes citrate (Fig. S18), cis-aconitate (Fig. S19), isocitrate (Fig. S13), alpha-ketoglutarate (Fig. S20), lactate (Fig. S21), and oxaloacetate (Fig. 4D). It is likely that the accumulation of group 3 organic acids also contributed to the decrease in pH value of the Δ*mtrA* growth medium at 48 h.

## MtrA targets carbon metabolism genes in *S. coelicolor*

Due to the broad impact of MtrA on carbon metabolism, we hypothesized that MtrA targets carbon metabolism genes. For a comprehensive identification of MtrA targets, the genome sequence of *S. coelicolor* was searched, using RegPredict (33) and MEME, for intergenic sites identical or nearly identical to the predicted consensus binding sequence of MtrA (5′-GTAACCNNNNNNGTAACC-3′) (12), and predicted sites that were upstream of carbon metabolism genes were selected (Table 1). Potential MtrA sites were revealed upstream of *dhsB* (*sco4855*), encoding a putative succinate dehydrogenase subunit; *sco4595*, encoding a putative 2-oxoglutarate oxidoreductase subunit; *citA* (*sco2736*), encoding a citrate synthase (26); and *sco4388* (*citA2*), encoding a putative citrate synthase; and also in the intergenic sequence between *sco5831* (*citA3*) and *sco5832* (*citA4*), both encoding putative citrate synthases. Next, we performed electrophoretic mobility shift assay (EMSA) analysis to determine if MtrA could bind short oligonucleotide probes containing the predicted MtrA sites. Our analysis showed that MtrA bound the probes with the predicted MtrA sites for *citA* and *citA2*, and the site

**TABLE 1** Potential MtrA sites upstream of carbon metabolism pathway genes in *S. coelicolor*[c]

| Gene | Function | Predicted MtrA site[a] | Position to TSS[b] |
|---|---|---|---|
| *SCO4855* (*dhsB*) | Putative succinate dehydrogenase iron-sulfur subunit | **GTCGC**GCGGCG**GTCAC** | −152 |
| *SCO4595* | 2-Oxoglutarate oxidoreductase, alpha subunit | **GTCAC**GGTGCG**CGGAC** | −71 |
| *SCO2736* | Citrate synthase | **GTCAC**ACAGCA**CTTCC** | −161 |
| *SCO4388* | Citrate synthase | **GTCAC**GCCGCC**CATCC** | −277 |
| *SCO5831* | Citrate synthase | **TTGAC**TTAACT**GTCCA** | −92 |
| *SCO5832* | Citrate synthase | | −16 |
| *SCO3817* (*bkdA1*) | Putative branched-chain alpha-keto acid dehydrogenase E1 alpha subunit keto acid dehydrogenase E1 alpha subunit | **GCGAA**CAGCAC**GTTAC** | −175 −132 |
| *SCO3818* | Putative two-component system response transcriptional regulator | | |
| *SCO3829* (*bkdC2*) | Putative dihydrolipoamide acyltransferase component E2 | **GTGAC**GCTCCA**GCATC** | −65 |
| *SCO3830* (*bkdB2*) | Putative branched-chain alpha-keto acid dehydrogenase E1 beta subunit | **GCGCG**GAGCTG**GTCCC** | −63 |
| *SCO3831* (*bkdA2*) | E1-alpha branched-chain alpha-keto acid dehydrogenase keto acid dehydrogenase | **GTCAC**ACCCGT**GGCGA** | −33 |
| *SCO3832* (*bkdR*) | Putative transcriptional regulator | | −246 |

[a]The potential MtrA sites were predicted using MEME software.
[b]TSS, translational start site.
[c]Boldface indicates the predicted MtrA recognition sequence of each gene.

between *citA3* and *citA4* (Fig. 6A). We then performed real-time PCR to determine if MtrA regulates these key carbon metabolism genes, using RNA extracted from 24 to 84 h, at intervals of 12 h, from cultures grown on YBP (Fig. 6B). Our transcriptional analysis showed differing patterns of expression for *citA* and *citA2* in M145, with *citA* showing the highest expression early, whereas the highest expression of *citA2* occurred late in the time course. For each gene, though, at the times of peak expression in M145, the expression was noticeably lower in the mutant, with the expression level of *citA* about twofold lower at the first two time points (24 and 36 h) in Δ*mtrA*, and the expression level of *citA2* markedly lower at the last two time points (72 and 84 h) in Δ*mtrA*, especially at 84 h, when the expression level was about one-tenth of that in M145. Whereas *citA3* showed a relatively consistent level of expression in M145, the pattern of *citA4* expression in M145 was more similar to that of *citA*. However, in contrast to *citA* and *citA2*, both *citA3* and *citA4* were expressed at notably higher levels in Δ*mtrA* than in M145 at late time points. Our analysis also showed that *sco4595*, which encodes a putative 2-oxoglutarate oxidoreductase, had markedly lower expression at later time points in Δ*mtrA* than in M145, a pattern similar to that of *citA2*, whereas *sdhB* had notably lower expression at the first two time points in Δ*mtrA* compared to M145, which is similar to *citA*. Our data demonstrated a positive regulatory effect of MtrA on *citA* at the early growth phase and on *citA2* at the late growth phase, but a generally negative regulatory effect of MtrA on *citA3* and *citA4*. Collectively, our data showed that MtrA targets and regulates carbon metabolism genes, although with some temporal variation in the effects of MtrA on these genes. Additionally, although we focused on examining the expression of carbon metabolism genes that had predicted MtrA sites in their upstream regions, other carbon metabolism genes may also be directly or indirectly regulated by MtrA. MtrA binding sites that did not closely match the consensus binding site have been reported (12), so it is possible that other carbon metabolism genes may be direct targets of MtrA. Furthermore, MtrA is known to regulate other transcription factors (9, 10, 12), and therefore, the effects of MtrA on carbon metabolism genes could be amplified beyond direct targets via other regulatory factors.

To investigate the functional conservation of MtrA in carbon metabolism, we searched for potential MtrA sites upstream of carbon metabolism genes in other *Streptomyces* species. Potential MtrA sites were predicted upstream of genes encoding citrate synthase, succinate dehydrogenase, 2-oxoglutarate oxidoreductase, and other proteins potentially involved in the TCA cycle in *S. lividans* (Table S1), *S. venezuelae*

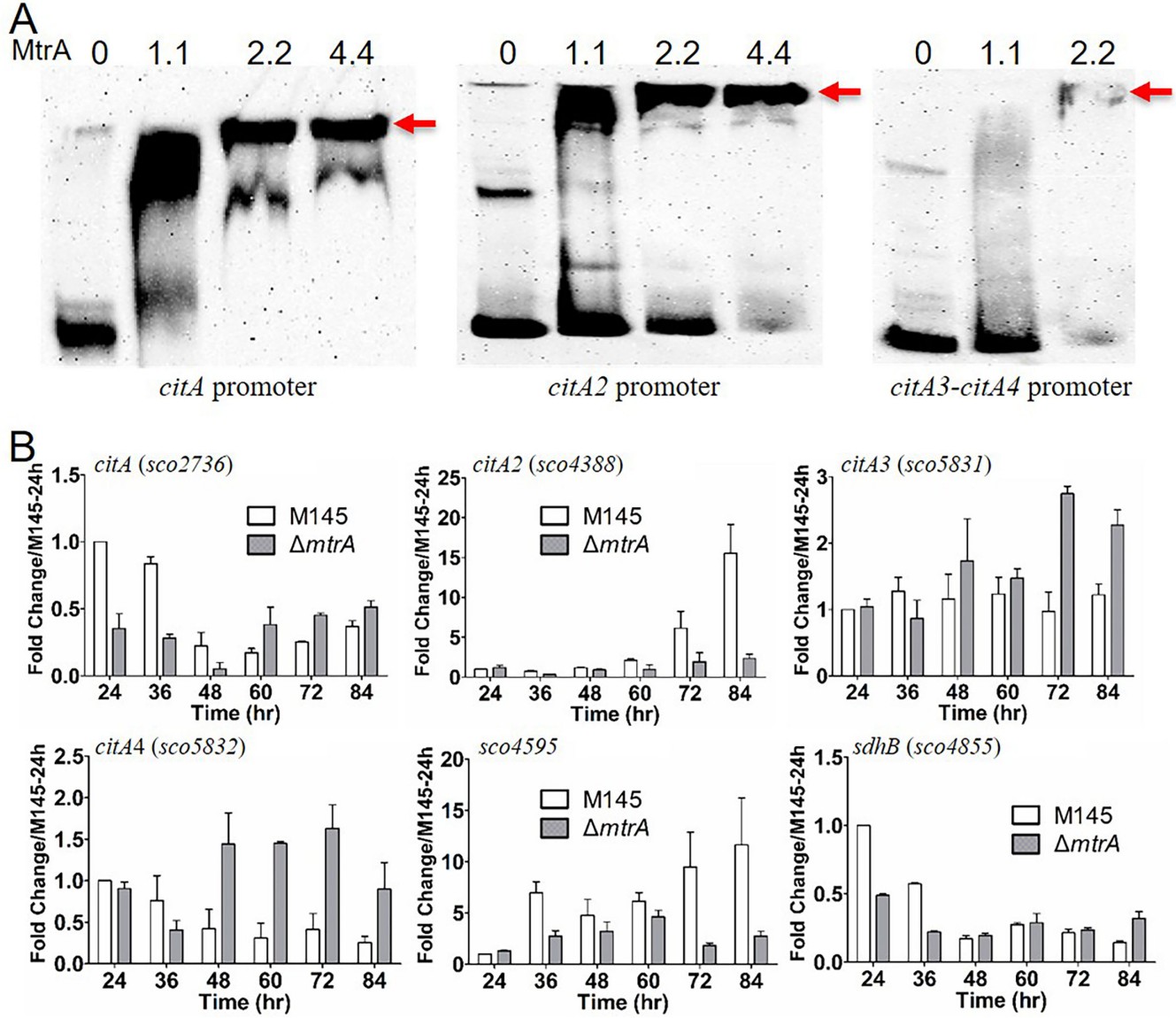

**FIG 6** EMSAs and transcriptional analysis of *S. coelicolor* genes with potential upstream MtrA recognition sequences. (A) EMSAs of MtrA with predicted sites similar to the MtrA consensus sequence. The probe sequences are located upstream of the named genes. A fixed amount of labeled oligonucleotide probe was incubated in reactions containing no MtrA (lane 1) or 1.1, 2.2, or 4.4 µg MtrA (lanes 2, 3, and 4, respectively). The red arrow indicates the shifted probe. (B) Transcriptional analysis of potential MtrA target genes in the Δ*mtrA* mutant. M145 and Δ*mtrA* were grown on YBP solid medium, and RNA samples were isolated at the indicated times. Expression of *hrdB,* encoding the major sigma factor in M145, was used as an internal control. The y-axis shows the fold change in expression level in M145 (light bars) over the level in Δ*mtrA* (gray bars) at each time point, with the expression level of each gene in M145 at 24 h arbitrarily set to one. Results are the means (±SD) of triplet biological experiments.

(Table S2), and *S. avermitilis* (Table S3). Similarly, MtrA sites were also predicted upstream of carbon metabolism genes in *Mycobacterium tuberculosis* (Table S4), *Amycolatopsis mediterranei* (Table S5), *Corynebacterium glutamicum* (Table S6), and *Mycobacterium smegmatis* (Table S7). The presence of potential MtrA sites upstream of carbon metabolism genes in these different actinobacterial species suggests a general role for MtrA in the regulation of carbon metabolism in these species and potentially in other actinobacteria.

Other regulators, including TCSs, are reported to have roles in the regulation of carbon metabolism. In addition to its role as the major nitrogen regulator (15, 16), GlnR was found to control the uptake and utilization of non-phosphotransferase-system

carbon sources, including maltose, sorbitol, mannitol, and mannose, in the actinomycete *Saccharopolyspora erythraea* (34), and potentially in other actinomycetes such as *Streptomyces* (34). Additionally, GlnR regulated three genes encoding citrate synthases, in collaboration with DasR and CRP, in *Saccharopolyspora erythraea* (34). Due to the similar sequences recognized by GlnR and MtrA (14), there may be some overlap in the control of carbon metabolism genes by these two major regulators. Moreover, studies indicate that the TCSs PhoR/PhoP, SCO5282/SCO5283, and RspA1/RspA2 also have roles in the regulation of central carbon metabolism in *Streptomyces* species (35–37), suggesting that complex processes are involved in controlling this key aspect of primary metabolism.

## Conclusion

In this study, we revealed that the response regulator MtrA has a broad impact on carbon metabolism, including the TCA cycle and glycolysis pathway, with marked changes in the production of carbon metabolism metabolites, especially organic acids, in the *S. coelicolor mtrA* mutant strain Δ*mtrA*, leading to the accumulation of multiple organic acids and the resulting acidification of the growth medium. Furthermore, we showed that MtrA interacts with predicted MtrA binding sites upstream of key carbon metabolism genes, such as *citA*, and we also demonstrated that these genes were differentially expressed in Δ*mtrA*, suggesting that MtrA targets and regulates carbon metabolism genes. Therefore, our findings extend the role of MtrA to regulation of carbon metabolism, in addition to its previously revealed roles as a regulator in developmental control (10, 12), antibiotic production (9–11), phosphate metabolism (13), and nitrogen metabolism (14, 17). These findings also reinforce the role of MtrA as a major regulator in *Streptomyces* metabolism.

## MATERIALS AND METHODS

### Culture conditions

*S. coelicolor* M145 was used as the wild-type strain, Δ*mtrA* is a deletion mutant of the response regulator genes *mtrA* in M145, and C-Δ*mtrA* is the complemented strain (12). YBP (2 g yeast extract, 2 g beef extract, 4 g Bacto peptone, 1 g $MgSO_4$, 15 g NaCl, 10 g glucose, and 20 g agar in 1 L $H_2O$, pH 7.0) agar was used for measurement of pH, determination of organic acids, and extraction of RNA (25). R2YE (R2 with the addition of yeast extract [0.5% final concentration]) was used for measurement of pH and determination of organic acids (24). R2 (103 g sucrose; 0.25 g $K_2SO_4$; 10.12 g $MgCl_2.6H_2O$; 10 g glucose; 0.1 g Difco casamino acids; 10 mL $KH_2PO_4$ [0.5%]; 80 mL $CaCl_2.2H_2O$ [3.68%]; 15 mL L-proline [20%]; 100 mL TES buffer [5.73%, adjusted to pH 7.2]; 2 mL trace element solution; and 20 g agar in 1 L $H_2O$, pH 7.0) and modified R2 (proline was replaced by either peptone [R2-proline + peptone] or tryptone [R2-proline + tryptone]) were used for measurement of pH (24). Equal amounts of spores (about $2 \times 10^6$) for *S. coelicolor* strains M145, Δ*mtrA*, and C-Δ*mtrA* were spread evenly on agar media and were grown at 30°C for the intended time before further treatment.

### Measurement of pH

Equal numbers (about $2 \times 10^6$) of *Streptomyces* spores for M145, Δ*mtrA*, and C-Δ*mtrA* were inoculated onto YBP and other media (R2YE, R2, and modified R2) and were incubated at 30°C for different times. For all experiments, after culturing the strains, the solid growth agar was weighed first and then cut into small pieces, and then an equal weight of distilled water was added, followed by vortexing for 2 h and centrifugation to remove the debris. The pH levels were then measured using a pH meter (Sartorius), essentially as described (38).

## Targeted metabolomics analysis

Frozen scraped cultures were thawed at 4°C, and 80 mg of each sample was mixed with 1 mL of cold methanol/acetonitrile/H$_2$O (2:2:1, vol/vol/vol) and 10 µL of isotope internal standards. The samples were homogenized using an MP homogenizer (20 s, thrice), vortexed, and then subjected to ultrasonication for 30 min on ice, followed by incubation at −20°C for 1 h. The lysates were then centrifuged for 20 min (14,000 RCF, 4°C), and the supernatants were dried in a vacuum centrifuge. For liquid chromatography-mass spectrometry (LC-MS) analysis, the samples were resuspended in 100 µL acetonitrile/water (1:1, vol/vol), vortexed, and then centrifuged (14,000 RCF, 4°C, 15 min). The supernatants were then collected for LC-MS/MS analysis, which was performed using an ultra-high performance liquid chromatography (1290 Infinity LC, Agilent Technologies) coupled to a QTRAP (AB Sciex 6500+). The mobile phase contained solvent A (50 mM CH$_3$COONH$_4$ in water and 1.2% NH$_4$OH) and B (1% acetylacetone in CH$_3$CN). The samples were loaded in the automatic sampler at 4°C, and the column temperature was kept constant at 35°C. The gradients were run at a flow rate of 300 µL/min, and 2 µL aliquots of each sample were injected. The gradient was 70% B for 0–1 min, which was then reduced to 60% B for 1–10 min, 30% B for 10–12 min, and then after another 12–15 min, B was increased to 70% for 15 min and kept for 15–22 min. Quality control (QC) samples were used for testing and evaluating the stability and repeatability of this system at the same time. A set of standard mixtures of metabolites was used for the correction of chromatographic retention time. MS/MS analysis using multiple reaction monitoring (MRM) was performed in ESI negative modes, and the conditions were set as follows: source temperature, 450°C; ion source gas 1 (Gas 1), 45; ion source gas 2 (Gas 2), 45; curtain gas, 30; ionspray voltage floating, −4,500 V; with the MRM mode detection ion pair. The MultiQuant software was used to extract the chromatographic peak areas and retention times. The correct retention time, based on the standards, was used to identify the metabolites.

## Electrophoretic mobility shift assays

For DNA probes containing putative MtrA binding sequences, oligonucleotides were labeled separately with biotin-11-UTP, and complementary oligonucleotide strands were annealed to generate probes for EMSAs. In the reactions, 50 fmol of labeled probes were mixed with different amounts of purified MtrA in binding buffer (20 mM Tris-HCl, 2 mM EDTA, 20 mM KCl, 0.5 mM DTT, and 4% Ficoll-400, pH 8.0), incubated at room temperature for 15 min, and then were separated on 8% non-denaturing polyacrylamide gels. After gel separation, the DNA was transferred to and fixed on nylon membranes, and then blocked, washed, and processed using standard procedures. Finally, the signal was detected using the ECL Western Blotting Analysis System kit (GE Healthcare), followed by exposure to X-ray film.

## RNA isolation, reverse transcription-PCR, and real-time PCR

To extract RNA, equal numbers (about 2 × 10$^6$) of *Streptomyces* spores for wild-type and mutant strains were grown at 30°C on YBP solid medium, covered with cellophane, and the mycelia were collected at multiple times, including 24, 36, 48, 60, 72, and 84 h, with 12 h intervals. Collected mycelia were ground in liquid nitrogen and then dispensed into REZol reagent (SBSBIO). Chloroform was added to the mixture, and the mixture was then vortexed for precipitation of cellular proteins. The supernatant was transferred to RNase-free Eppendorf tubes after centrifugation for 10 min at high speed; absolute ethanol was then added, the samples were mixed, and then transferred to absorption columns. The columns were centrifuged to remove the liquid, washed twice with washing buffer, followed by two elutions with DEPC-water; the eluates were then collected, and crude RNA was precipitated with the addition of absolute ethanol and sodium acetate. Crude RNA samples were treated twice with 'Turbo DNA-free' DNase reagents (Ambion) to remove chromosomal DNA, and reverse transcription was carried

out, as described (12). SYBR Green *Pro Taq* HS (AG Bio) was used under recommended conditions on a Roche LightCycler 480 thermal cycler to determine the melting curve of PCR products and their specificity, and for real-time PCR assays. Relative quantities of cDNA were normalized using the *hrdB* gene, which encodes the major sigma factor of *Streptomyces*, and the expression level of each gene in M145 at 24 h was arbitrarily set to one.

## Construction of an *mtrA* expression plasmid and purification of MtrA protein

The *mtrA* coding sequence was amplified using primers *mtrA*-Exp-F (with an *Nde*I adaptor, 5′-CGCCATATGATGTCGTTTATGAAGGGACGAG-3′) and *mtrA*-Exp-R (with a *Hind*III adaptor, 5′-CCCAAGCTTTCAGCTCGGTCCGGCCTTGTAGCCG-3′), and the PCR product was purified by agarose gel electrophoresis and inserted into pMD18-T (Takara). After sequence verification, the inserts were excised by *Nde*I and *Hind*III digestion, gel-purified, and ligated into *Nde*I/*Hind*III-cut pET28a (Invitrogen) to generate pEX-*mtrA*, which was used to transform *Escherichia coli* Rosetta(DE3)pLysS (Novagen). Expression of MtrA was induced by the addition of isopropyl β-D-1-thiogalactopyranoside (1.0 mM) when the cell density was around 0.6 (at $OD_{600\ nm}$), with incubation for 4 to 5 h at 30°C. Cell lysates were prepared by sonication in binding buffer (50 mM $NaH_2PO_4$, 250 mM NaCl, 20 mM imidazole, pH 8.0), and the His-tagged MtrA was purified using Ni-NTA-Sefinose Column (Sangon), using washing buffer (50 mM $NaH_2PO_4$, 250 mM NaCl, 40 mM imidazole, pH 8.0) and then elution buffer (50 mM $NaH_2PO_4$, 250 mM NaCl, 250 mM imidazole, pH 8.0). Purified protein was then dialyzed in dialysis cassettes (10,000 MWCO, Thermo Scientific) in a dialyzing buffer (50 mM $NaH_2PO_4$, 50 mM NaCl, pH 8.0) before concentrating with centrifugal filters (10,000 MWCO, Millipore). Protein concentration was determined using the bicinchoninic acid assay (Pierce).

## Scanning electron microscopy

Briefly, spores of *S. coelicolor* M145 and its derivative strains were inoculated onto MS agar medium; for *S. venezuelae* strains, YBP agar medium was used instead. A sterile coverslip was inserted into the agar at an angle to allow the culture to overgrow its surface. After 5 days of incubation at 30°C, the coverslip was removed, fixed with fresh 2% glutaraldehyde (pH 7.2) for 2 h at 30°C, and washed three times with 0.1 M PBS buffer (pH 7.0) before being treated with 1% osmic acid. The coverslips were then dehydrated by soaking in a series of ethanol gradients, dried in a Leica EM CPD300 Critical Point Dryer, coated with gold in a Cressington Sputter Coater 108, and examined with a scanning electron microscope (FEI Quanta250 FEG, USA).

## ACKNOWLEDGMENTS

This work was supported by grants from the National Natural Science Foundation of China (NSF32270072 to X.P. and NSF32300046 to Y.Z.), the Open Funding Project of State Key Laboratory of Microbial Metabolism (MMKF24-04 to X.P.), the Basic Research Program of Jiangsu (BK20230247 to Y.Z.), the Natural Science Foundation of Shandong Province (ZR2023QC172 to Y.Z.), and the Qingdao Natural Science Foundation (23-2-1-27-zyyd-jch to Y.Z.).

## AUTHOR AFFILIATIONS

[1]The State Key Laboratory of Microbial Technology, Shandong University, Qingdao, Shandong, China
[2]Suzhou Research Institute, Shandong University, Suzhou, Jiangsu, China

## AUTHOR ORCIDs

Yanping Zhu http://orcid.org/0000-0002-2955-6585
Xiuhua Pang http://orcid.org/0000-0002-0115-1973

## AUTHOR CONTRIBUTIONS

Yanping Zhu, Formal analysis, Funding acquisition, Investigation | Hanlei Zhang, Investigation | Xiuhua Pang, Conceptualization, Formal analysis, Funding acquisition, Supervision, Writing – original draft, Writing – review and editing

## ADDITIONAL FILES

The following material is available online.

### Supplemental Material

**Supplemental material (Spectrum00096-25-s0001.pdf).** Fig. S1 to S21; Tables S1 to S7.

### Open Peer Review

**PEER REVIEW HISTORY (review-history.pdf).** An accounting of the reviewer comments and feedback.

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
