## [Reviewer comments · Microbiology Spectrum]

Microbiology Spectrum

Metabolomic analysis of the impact of MtrA on carbon metabolism in *Streptomyces coelicolor*

Yanping Zhu, Hanlei Zhang, and Xiuhua Pang

Corresponding Author(s): Xiuhua Pang, Shandong University - Qingdao Campus

Review Timeline:

Submission Date:	January 9, 2025
Editorial Decision:	February 8, 2025
Revision Received:	March 25, 2025
Editorial Decision:	April 16, 2025
Revision Received:	May 6, 2025
Editorial Decision:	May 22, 2025
Revision Received:	May 22, 2025
Accepted:	May 23, 2025

Editor: Silvia Cardona

Reviewer(s): Disclosure of reviewer identity is with reference to reviewer comments included in decision letter(s). The following individuals involved in review of your submission have agreed to reveal their identity: Guoqing Niu (Reviewer #1)

Transaction Report:

DOI: <https://doi.org/10.1128/spectrum.00096-25>

Re: Spectrum00096-25 (Metabolomics analysis of the impact of MtrA on carbon metabolism in *Streptomyces coelicolor*)

Dear Dr. Xiuhua Pang:

Thank you for submitting your manuscript to Microbiology Spectrum. Your article has been reviewed by two experts in the field. Reviewers found the research design to be sound; however there are concerns that need to be addressed before the article is considered for publication. I concur with the reviewers that the figures need work to improve clarity and some results (several media) are repetitive. In addition, consider internal review of the metabolomics analysis by a colleague with more expertise in the field so that the more relevant results are presented.

Their recommendations are provided below.

Revision Guidelines

Sincerely,
Silvia Cardona
Editor
Microbiology Spectrum

Reviewer #1 (Comments for the Author):

This study by Zhu and colleagues employed targeted metabolomics to compare the organic acids produced by the model organism *S. coelicolor* M145 and its MtrA mutant strain. Significant alterations were detected in the production of multiple organic acids within the TCA cycle and glycolysis pathway in mtrA mutant. These findings indicated that MtrA exerts a substantial influence on carbon metabolism and the acidification of the cultivation media. Bioinformatics and EMSAs identified MtrA binding sites situated upstream of genes related to carbon metabolism. Furthermore, transcriptional levels of genes in carbon metabolism was also modified in mtrA mutant. Taken together, this study demonstrated that the response regulator MtrA has a profound impact on central carbon metabolism of *Streptomyces*. Overall, this study is well - designed, and all experiments have been properly executed. However, there are several concerns that require the authors' attention. Specific comments are as follows:

1. The introduction is rather concise. It should be expanded to include a concise description of the current regulatory network of mtrA and regulators belonging to the same family. It is also advisable to include a brief introduction of central carbon metabolism in *Streptomyces*.
2. The layout of the figures, along with their labelling, are rather confusing. For example, parts of the Fig. 1A and 2A are exactly the same. Fig. 1C can be supplied as a separated figure in the supplementary materials. Figure 2A should be incorporated into Fig 1.
3. There are several figures showing the growth of strains on different cultivation media. However, microscopic observations of the strains grown on these media are lacking. My suggestion is include light microscopy and/or SEM, at least with these strains cultivated on selected growth media. Moreover, no results related to Fig S1 were presented in the Results section.
4. Line 158. Please specify why YBP and R2YE were selected here.
5. Line 248. Please provide the full sequence of consensus binding sequence of MtrA in the main text.
6. Materials and Methods. This section is rather concise. Please provide more detailed information for "Culture conditions", "Measurement of pH" and "RNA isolation, reverse transcription PCR (RT PCR), and real time PCR". The overexpression of MtrA should also be included in this section.

Reviewer #2 (Comments for the Author):

This manuscript deals with the possible regulatory role of MtrA on some genes of carbon metabolism.

Line 110. Three components of the YBP medium are mentioned, however, one very important one is omitted: glucose, which when metabolized through the glycolytic pathway and the tricarboxylic acid cycle, generates organic acids that can cause drops in the pH value. Therefore, the results of acidification of the *S. coelicolor* growth medium are sufficient to be able to say that something is happening in the mtrA deleted mutant.

It is considered that the results with the MS, R2YE and R2 media provide information that is not so relevant and that excluding them reduces the number of figures and allows a better and more interesting analysis of the results.

It should be clarified that yeast extract being a mixture of proteins and carbohydrates, among other components, are carbohydrates and some amino acids that cause the decrease in pH.

Proline is a glucogenic amino acid since when metabolized it generates glutamate and by the action of glutamate dehydrogenase it forms α -ketoglutarate, an intermediate of the tricarboxylic acid cycle, which would explain the acidification of the medium. Therefore, the decrease in pH in the medium is explainable and it is not necessary to present results in different media.

The same results obtained with the mtrA deleted mutants in *Streptomyces lividans* and *Streptomyces venezuelae* could only be presented in supplementary information.

The paragraph (lines 169-173) includes very risky assertions that are not verified, so it is better to avoid them.

What is the difference between the growth and metabolomics described in the second paragraph on page 7 and the last paragraph (lines 208-221) on the same page? It seems to me that instead of being R2YE in the last paragraph the medium is R2. Still, I think these results can be omitted since there is no additional relevant information.

It is not clear why it is considered important to classify the levels of the generated organic acids into 3 groups, so the paragraph (lines 227-242) can be omitted.

The results of the real-time PCR show that the mRNA levels of the sco2736 and sco4388 genes encoding citrate synthase in the mtrA-deleted mutant are lower than in the wild-type strain. The mRNA levels of sco5831 and sco5832 at times 48-72 are higher in the mutant. It is not clear why it is claimed that the probable high activity of the citrate synthase encoded by these two genes is responsible for the accumulation of citrate, if at these times the pH value tends to rise.

The absence of MtrA causes the decrease of the mRNA levels of sco4595 and sco4855 at some times but does not show a clear regulatory pattern on these genes. In this manuscript they demonstrate that MtrA can have a positive regulatory role in the expression of the genes sco2736, sco4388 and sco4595 while it could have a negative effect, at some times, on the genes sco5831 and sco5832, however, a major modification is required.

Response to comments

We thank the editor and reviewers for their helpful comments. As indicated below, we have addressed the specific concerns, including making changes to the figures, adding more details to the introduction and methods, and other requested revisions.

Editor's comment: ...consider internal review of the metabolomics analysis by a colleague with more expertise in the field so that the more relevant results are presented

Reply: Our metabolomics results have been rechecked by one of our coauthors who has considerable expertise in this field, and we believe that we have presented the most important results. However, if there are specific results that you would like to see added or removed, we are happy to make those changes.

Reviewer #1 (Comments for the Author):

Q1: The introduction is rather concise. It should be expanded to include a concise description of the current regulatory network of *mtrA* and regulators belonging to the same family. It is also advisable to include a brief introduction of central carbon metabolism in *Streptomyces*.

Reply: The introduction was expanded from 29 lines in the previous version to 51 lines in the current version, with background information added about MtrA and other regulators as well as carbon metabolism, as suggested.

Q2: The layout of the figures, along with their labelling, are rather confusing. For example, parts of the Fig. 1A and 2A are exactly the same. Fig. 1C can be supplied as a separated figure in the supplementary materials. Figure 2A should be incorporated into Fig 1.

Reply: Figure 1 and Figure 2 of the previous submission were combined into one figure (Figure 1 of current version), and the rest of the panels were moved to Supplementary Materials, as suggested.

Q3: There are several figures showing the growth of strains on different cultivation media. However, microscopic observations of the strains grown on these media are lacking. My suggestion is include light microscopy and/or SEM, at least with these strains cultivated on selected growth media. Moreover, no results related to Fig S1 were presented in the Results section.

Reply: Please note that our group already published images showing the morphology of these strains grown on YBP (see reference 12). For the morphology of the wild-type *S. coelicolor* strain M145 and the mutant strain $\Delta mtrA$ grown on R2YE and R2, please see the newly added Fig. S3B and Fig. S4B, respectively. For results related to Figure S1, please see lines 117 and 169.

Q4: Line 158. Please specify why YBP and R2YE were selected here.

Reply: YBP and R2YE were selected since they were acidified by $\Delta mtrA$, and this explanation has been added (lines 182-183).

Q5: Line 248. Please provide the full sequence of consensus binding sequence of MtrA in the main text.

Reply: The full consensus sequence for MtrA binding was provided as recommended (line 263).

Q6: Materials and Methods. This section is rather concise. Please provide more detailed information for "Culture conditions", "Measurement of pH" and "RNA isolation, reverse transcription PCR (RT PCR), and real time PCR". The overexpression of MtrA should also be included in this section.

Reply: Detailed information was added, as recommended, expanding this section from 59 lines in the previous submission to 104 lines in the current version (e.g., lines 319-330, 377-397, 395-411, and 413-421).

Reviewer #2 (Comments for the Author):

Q1: Line 110. Three components of the YBP medium are mentioned, however, one very important one is omitted: glucose, which when metabolized through the glycolytic pathway and the tricarboxylic acid cycle, generates organic acids that can cause drops in the pH value. Therefore, the results of acidification of the *S. coelicolor* growth medium are sufficient to be able to say that something is happening in the *mtrA* deleted mutant.

Reply: The missed glucose was added in line 132, and the detailed recipe for YBP was provided in lines 319-321. We agree with the conclusion that acidification of the growth medium by the *mtrA* mutant indicates that something is happening with this strain.

Q2: It is considered that the results with the MS, R2YE and R2 media provide information that is not so relevant and that excluding them reduces the number of figures and allows a better and more interesting analysis of the results.

Reply: We provided data from these media to show that acidification could occur under multiple growth conditions. We believe that it is better to keep these data to provide a broader view about the ability of the *mtrA* mutant to acidify growth media under different growth conditions.

Q3: It should be clarified that yeast extract being a mixture of proteins and carbohydrates, among other components, are carbohydrates and some amino acids that cause the decrease in pH.

Reply: This information was provided as suggested (lines 142-143).

Q4: Proline is a glucogenic amino acid since when metabolized it generates glutamate and by the action of glutamate dehydrogenase it forms α -ketoglutarate, an intermediate of the tricarboxylic acid cycle, which would explain the acidification of the medium. Therefore, the decrease in pH in the medium is explainable and it is not necessary to present results in different media.

Reply: Metabolism of the glucogenic amino acid proline (and other components such as yeast) could potentially lead to acidification of the growth medium, as deduced above, and this information was added as suggested (lines 148-150). However, please note that it is the absence of MtrA (the *mtrA* mutant) that makes the difference in pH on medium with proline such as R2 (Figure 1), while the wild-type and complemented strains do not produce a pH change under the same growth condition. As we tested the effect of MtrA on acidification of growth media under multiple other growth conditions, therefore we believe that it is better to present all of these results for a more comprehensive comparison of the effects of *mtrA* deletion.

Q5: The same results obtained with the *mtrA* deleted mutants in *Streptomyces lividans* and *Streptomyces venezuelae* could only be presented in supplementary information.

Reply: These data were removed from Fig. 2 of previous submission and are now presented in supplementary figure 8.

Q6: The paragraph (lines 169-173) includes very risky assertions that are not verified, so it is better to avoid them.

Reply: This paragraph was deleted as suggested.

Q7: What is the difference between the growth and metabolomics described in the second paragraph on page 7 and the last paragraph (lines 208-221) on the same page? It seems to me that instead of being R2YE in the last paragraph the medium is R2. Still, I think these results can be omitted since there is no additional relevant information.

Reply: The second paragraph described the growth and metabolomics data for 36 h on R2YE, while the last paragraph (lines 208-221) described the growth and metabolomics data for 48 h on R2YE, so that a dynamic pattern could be observed for the metabolites. However, we have shortened this section by several lines to focus on the main differences in patterns between the two time points (lines 230-237).

Q8: It is not clear why it is considered important to classify the levels of the generated organic acids into 3 groups, so the paragraph (lines 227-242) can be omitted.

Reply: We believed it would be easier to understand these data by classifying the organic acids into different groups according to the pattern of change. These groupings highlighted the variation in patterns among the organic acids and also allowed us to clarify how we identified the organic acids that likely contributed to the reduction in pH, and therefore we believe that it is very helpful to keep this information. However, this section was reduced.

Q9: The results of the real-time PCR show that the mRNA levels of the *sco2736* and *sco4388* genes encoding citrate synthase in the *mtrA*-deleted mutant are lower than in the wild-type strain. The mRNA levels of *sco5831* and *sco5832* at times 48-72 are higher in the mutant. It is not clear why it is claimed that the probable high activity of the citrate synthase encoded by these two genes is responsible for the accumulation of citrate, if at these times the pH value tends to rise.

Reply: The statement was removed to avoid misunderstanding.

Q10: The absence of MtrA causes the decrease of the mRNA levels of *sco4595* and *sco4855* at some times but does not show a clear regulatory pattern on these genes. In this manuscript they demonstrate that MtrA can have a positive regulatory role in the expression of the genes *sco2736*, *sco4388* and *sco4595* while it could have a negative effect, at some times, on the genes *sco5831* and *sco5832*, however, a major modification is required.

Reply: Revised as suggested (lines 287-289).

Re: Spectrum00096-25R1 (Metabolomics analysis of the impact of MtrA on carbon metabolism in *Streptomyces coelicolor*)

Dear Dr. Xiuhua Pang:

Your manuscript has improved. One reviewer still wants some clarification, which you will find below.

Revision Guidelines

Sincerely,
Silvia Cardona
Editor
Microbiology Spectrum

Reviewer #1 (Comments for the Author):

I have no further comments.

Reviewer #2 (Comments for the Author):

The manuscript has been greatly improved in the introduction, results, and conclusions chapters, but the results are not discussed nor are the results obtained compared with other published articles.

Lane 134, add "extract" after yeast.

In the results and discussion section, only the results are described, but the reasons for this, for example, the decrease in pH, are not discussed. For example, paragraphs 142-153 present the results of medium acidification in various complex media with a protein and peptide content of around 50%, and no suggestion is made as to why the metabolism of the medium components might cause the pH drop. In paragraphs 157-162 it is indicated that MtrA has a role as a repressor of nitrogen metabolism and that the deleted mutant of the *glnR* gene does not acidify the medium because GlnR has a role as an activator of nitrogen metabolism in a nutrient-poor medium, but the media mentioned are rich media.

Line 163 states that the growth of microorganisms is promoted at alkaline pH values, which is not true, since microorganisms grow preferentially at neutral pH. Include references to confirm these statements.

It is also extrapolated that the intracellular accumulation of different organic acids is the cause of the acidification of the culture medium, for which they have to be secreted.

Lane 220, page 8 change (Fig. 3) to (Figs. 2 and 3) as they describe the results shown in these two figures.

Change ";" to "," on lane 224.

Lane 228, page "we deduced that the accumulation of oxaloacetate led to the acidic pH at this point", however, lane 256 describes

"It is likely that accumulation of group 3 organic acids led to the decrease in pH value of the mutant deleted *mtrA* growth medium at 48 h". This seems to be contradictory.

Lane 236, It is commented that "potentially due to the greater expression of lactate dehydrogenase", however, according to KEGG *S. coelicolor* does not have a gene that codes for this enzyme (<https://www.kegg.jp/pathway/sco00010+SCO1947>).

The repressive or activating effect of genes encoding citrate synthases, a probable 2-oxoglutarate dehydrogenase, and one of the succinate dehydrogenase subunits, varies depending on the time at which the mRNA was quantified. This is inconsistent with the claim that MtrA regulates carbon metabolism genes.

It is assumed that malate dehydrogenase, pyruvate and phosphoenolpyruvate carboxylases are not regulated by MtrA, since they were not included in the study of gene expression, and they are the ones that generate oxaloacetate within the Krebs cycle and the phosphoenolpyruvate node.

In some of the figures change μ to μ .

These modifications are necessary for this manuscript to be published in this journal.

Response to reviewer's comments:

Reviewer #2 (Comments for the Author):

Q1. The manuscript has been greatly improved in the introduction, results, and conclusions chapters, but the results are not discussed nor are the results obtained compared with other published articles.

Reply: In the previous version, the results of this study were compared with some other studies (e.g., lines 170-174, 176-187). However, we agree that additional comparison would be helpful, and therefore, we have added more discussion (e.g., lines 153-158, 166-169, 175-176, 187-191, 249-251, 307-313, 324-334), as suggested.

Q2. Lane 134, add "extract" after yeast.

Reply: Added as suggested (line 134).

Q3. In the results and discussion section, only the results are described, but the reasons for this, for example, the decrease in pH, are not discussed. For example, paragraphs 142-153 present the results of medium acidification in various complex media with a protein and peptide content of around 50%, and no suggestion is made as to why the metabolism of the medium components might cause the pH drop. In paragraphs 157-162 it is indicated that MtrA has a role as a repressor of nitrogen metabolism and that the deleted mutant of the *glnR* gene does not acidify the medium because GlnR has a role as an activator of nitrogen metabolism in a nutrient-poor medium, but the media mentioned are rich media.

Reply: The potential reason why the metabolism of the medium components might cause the pH drop is discussed (lines 153-158, as suggested).

As noted, GlnR has a role as an activator of nitrogen metabolism in nutrient-poor medium but not under nutrient-rich medium, and we have clarified the corresponding information (lines 166-169).

Q4. Line 163 states that the growth of microorganisms is promoted at alkaline pH values, which is not true, since microorganisms grow preferentially at neutral pH. Include references to confirm these statements.

Reply: The statement was revised (line 170), and references were added, as suggested.

Q5. It is also extrapolated that the intracellular accumulation of different organic acids is the cause of the acidification of the culture medium, for which they have to be secreted.

Reply: Although the organic acids may accumulate intracellularly, it is true that, for the acidification of the culture medium, they would also need to be excreted. However, the excretion of organic acids has been reported for *Streptomyces*, and this information has been added and cited (lines 157-158).

Q6. Lane 220, page 8 change (Fig. 3) to (Figs. 2 and 3) as they describe the results shown in these two figures.

Change ";" to "," on lane 224.

Reply: These changes were made, as suggested (lines 222, 233, 237).

Q7. Lane 228, page "we deduced that the accumulation of oxaloacetate led to the acidic pH at this point", however, lane 256 describes "It is likely that accumulation of group 3 organic acids led to the decrease in pH value of the mutant deleted *mtrA* growth medium at 48 h". This seems to be contradictory.

Reply: We meant that the decrease in pH could result from the accumulation of oxaloacetate as well as

group 3 organic acids, so we have revised these statements to indicate that they “contributed” to the acidic pH (lines 241 and 272).

Q8. Lane 236, It is commented that "potentially due to the greater expression of lactate dehydrogenase", however, according to KEGG *S. coelicolor* does not have a gene that codes for this enzyme (<https://www.kegg.jp/pathway/sco00010>+SCO1947).

Reply: The product of SCO3594 is annotated as a putative D-lactate dehydrogenase; and we now indicate this possibility in the manuscript (lines 249-251).

Q9. The repressive or activating effect of genes encoding citrate synthases, a probable 2-oxoglutarate dehydrogenase, and one of the succinate dehydrogenase subunits, varies depending on the time at which the mRNA was quantified. This is inconsistent with the claim that MtrA regulates carbon metabolism genes.

It is assumed that malate dehydrogenase, pyruvate and phosphoenolpyruvate carboxylases are not regulated by MtrA, since they were not included in the study of gene expression, and they are the ones that generate oxaloacetate within the Krebs cycle and the phosphoenolpyruvate node.

Reply: The regulatory effect of genes encoding citrate synthases, a probable 2-oxoglutarate dehydrogenase, and one of the succinate dehydrogenase subunits varies depending on the time, suggesting a temporal regulatory effect of MtrA on these genes, which is still consistent with the conclusion that MtrA regulates carbon metabolism genes. However, the statement was revised (line 306).

Genes encoding citrate synthases, a probable 2-oxoglutarate dehydrogenase, and one of the succinate dehydrogenase subunits investigated in this study were selected based on the presence of MtrA sites upstream of these genes; however, that does not exclude the possibility that genes without predicted MtrA sites, such as those encoding malate dehydrogenase, pyruvate, and phosphoenolpyruvate carboxylases, are also regulated by MtrA. We have now discussed this possibility, lines 307-313.

Q10. In some of the figures change u to μ .

Reply: All u were changed to μ in Figs. 3, 4, and 5.

Re: Spectrum00096-25R2 (Metabolomics analysis of the impact of MtrA on carbon metabolism in *Streptomyces coelicolor*)

Dear Dr. Xiuhua Pang:

There are minor changes pending. Please address them so I can recommend acceptance.

Revision Guidelines

Sincerely,
Silvia Cardona
Editor
Microbiology Spectrum

Reviewer #2 (Comments for the Author):

The manuscript has improved significantly and is therefore acceptable for publication.

Only a few changes are recommended, as detailed below:

Page 5, lane 149. Change a- for α

Page 6, lane 170. Change "represses" for "negatively affected"

Page 6, lane 174. Change "mutation" for "deletion"

Page 6, lane 180. Change "repressed" for "poor growth"

Page 7, lane 213. Eliminate "s" de "metabolomics"
Page 17, lane 552. Change "PANS" for "PNAS"

Response to reviewer's comments:

Reviewer #2 (Comments for the Author):

Page 5, line 149. Change a- for α

Page 6, line 170. Change "represses" for "negatively affected"

Page 6, line 174. Change "mutation" for "deletion"

Page 6, line 180. Change "repressed" for "poor growth"

Page 7, line 213. Eliminate "s" de "metabolomics"

Page 17, line 552. Change "PANS" for "PNAS"

Reply: All recommended changes were corrected, as suggested.

Re: Spectrum00096-25R3 (Metabolomic analysis of the impact of MtrA on carbon metabolism in *Streptomyces coelicolor*)

Dear Dr. Xiuhua Pang:

Your manuscript has been accepted, and I am forwarding it to the ASM production staff for publication. Your paper will first be checked to make sure all elements meet the technical requirements. ASM staff will contact you if anything needs to be revised before copyediting and production can begin. Otherwise, you will be notified when your proofs are ready to be viewed.

Sincerely,
Silvia Cardona
Editor
Microbiology Spectrum